# Prevalence and socio-demographic correlates of accelerometer measured physical activity levels of school-going children in Kampala city, Uganda

Bernadette Nakabazzi[1,2]*, Lucy-Joy M. Wachira[1], Adewale L. Oyeyemi[3], Ronald Ssenyonga[4], Vincent O. Onywera[1]

1 Department of Physical Education, Exercise and Sports Science, School of Public Health and Applied Human Sciences, Kenyatta University, Nairobi, Kenya, 2 Department of Biochemistry and Sports Science, College of Natural Science, Makerere University, Kampala, Uganda, 3 Department of Physiotherapy, College of Medical Sciences, University of Maiduguri, Maiduguri, Nigeria, 4 Department of Epidemiology and Biostatistics, College of Health Sciences, Makerere University, Kampala, Uganda

* bnakabazzi@gmail.com

**Data Availability Statement:** All relevant data are within the manuscript and its Supporting Information files.

## Abstract

### Background

The current international physical activity guidelines for health recommend children to engage in at least 60 minutes of moderate-to-vigorous physical activity (MVPA) daily. Yet, accurate prevalence estimates of physical activity levels of children are unavailable in many African countries due to the dearth of accelerometer-measured physical activity data. The aim of this study was to describe the prevalence and examine the socio-demographic correlates of accelerometer-measured physical activity among school-going children in Kampala city, Uganda.

### Methods

A cross-sectional study design was used to recruit a sample of 10–12 years old school-going children (n = 256) from 7 primary schools (3 public schools and 4 private schools) in Kampala city, Uganda. Sedentary time, light-intensity physical activity (LPA), moderate-intensity physical activity (MPA) and vigorous-intensity physical activity (VPA) were measured by accelerometers (ActiGraph GT3X+ [Pensacola, Florida, USA]) over a seven-day period. Socio-demographic factors were assessed by a parent/guardian questionnaire. Weight status was generated from objectively measured height and weight and computed as body mass index (BMI). Multi-level logistic regressions identified socio-demographic factors that were associated with meeting physical activity guidelines.

### Results

Children's sedentary time was 9.8±2.1 hours/day and MVPA was 56±25.7 minutes/day. Only 36.3% of the children (38.9% boys, 34.3% girls) met the physical activity guidelines. Boys, thin/normal weight and public school children had significantly higher mean daily MVPA levels. Socio-demographic factors associated with odds of meeting physical activity

**Funding:** The PhD project was funded by the African Development Bank-Higher Education in Science and Technology (AfDB-HEST), Makerere University, Kampala

**Competing interests:** The authors have declared that no competing interest exists.

guidelines were younger age (OR = 0.68; 95% CI = 0.55–0.84), thin/normal weight status (OR = 4.08; 95% CI = 1.42–11.76), and socioeconomic status (SES) indicators such as lower maternal level of education (OR = 2.43; 95% CI = 1.84–3.21) and no family car (OR = 0.31; 95% CI = 0.17–0.55).

## Conclusion

Children spent a substantial amount of time sedentary and in LPA and less time in MVPA. Few children met the physical activity guidelines. Lower weight status, lower maternal education level and no family car were associated with meeting physical activity guidelines. Effective interventions and policies to increase physical activity among school-going children in Kampala, are urgently needed.

## Introduction

Childhood physical activity is associated with numerous physical, psycho-social and cognitive health benefits [1,2]. All levels of physical activity; LPA, MPA and VPA are important [2]. LPA contributes the most to overall physical activity and may be easier for children to engage in; however, higher physical activity intensity levels (MVPA) are consistently associated with greater health benefits [2,3]. In this regard, the World Health Organisation (WHO) [4] and some countries such as United States of America (USA) [5], Canada [6], Australia [7] and United Kingdom (UK) [8] have established and revised physical activity guidelines for children. The international physical activity guidelines recommend children to accumulate at least 60 minutes of MVPA each day to acquire the health benefits. However, even with known health benefits associated with regular participation in physical activity, global estimates show that 81% of children aged 11 to 17 years are not sufficiently active [9]. In 2016, Sallis and colleagues found no evidence of global increases in physical activity [10]. A recent study on global trends in insufficient physical activity among adolescents also found that 4 in every 5 adolescents aged 11 to 17 years did not meet the current physical activity guidelines [11]. Also across Europe, a harmonized analysis of accelerometer-measured physical activity revealed that two thirds of European children and adolescents were not sufficiently active [12]. The global pattern of insufficient physical activity in children has also been observed in Sub-Saharan African countries [13,14,15,16] particularly in urban areas. For example, in neighbouring Nairobi city results from the International Study of Childhood Obesity, Lifestyle and the Environment study (ISCOLE) showed that only 12.6% of the children 10 to 11 years old met the physical activity guidelines [17]. This is an indication that insufficient physical activity is still a current global public health problem. Findings from a recent systematic review and meta-analysis of longitudinal studies also showed that physical activity starts to decline in childhood [18]. Promoting physical activity during childhood is therefore a public health priority because this behavior persists into adolescence and adulthood [19]. However, a recent study on global trends in insufficient physical activity among children [11] and a publication on physical activity report cards from nine low-and middle-income countries (LMICs) [20], found a challenging data gap particularly in accelerometer-measured physical activity. Therefore, there is an urgent need for quality data to better describe children's physical activity and associated factors. Accurate measurement of children's physical activity is also key to continued surveillance and formulation of informed interventions and polices.

Technological advances in past two decades have increased the use of accelerometers to quantify children's physical activity [21]. Accelerometers are an alternative to self-reporting methods like questionnaires that are subject to recall bias and are not recommended for use among children because of their limited reading and comprehension skills due to their age [22,23]. Recall-based measures may also not accurately capture the sporadic and short-burst patterns of children's physical activity and LPA [24,25]. Accelerometers provide a valid and reliable measure of patterns as well as total physical activity among children [26,27]. Despite the increase in the use of accelerometers to quantify children's physical activity in large population studies, especially in high income countries (HICs) [28,29,30], there are relatively fewer studies that have used accelerometers in low income countries (LICs) [11,20]. Accelerometer-measured physical activity data was also non-existent in school-going children in Kampala city, Uganda. Theron and Santorino in 2009 used photographic methods to study physical activity of Ugandan youth in Mbarara and found that they participated in physical activity for 1 to 2 hours/week [31]. Analysis of self-reported secondary data from the Global School-based Students Health Survey (GSHS) showed that most Ugandan adolescents aged 13 to 15 years were inactive [32]. A pilot study among urban and rural school going children 10 to 16 years old from central Uganda, reported varied physical activity engaged in (such as active travel to school, sport-related, house chores and muscle-strengthening activities). However, the study used self-reporting measures [33]. Therefore, there was a need for accelerometer-measured physical activity data, to describe children's physical activity levels and identify the proportion of children who complied with the physical activity guidelines in Kampala city, Uganda.

Children's physical activity is consistently associated with various socio-demographic factors [29,34,35,36]. Various studies that used both objective and recall-based measures of physical activity consistently reported sex differences in physical activity favouring boys [11,17,29,34,35,36,37]. Children's physical activity has also been found to decline with increasing age [18,29], nevertheless non-significant associations have also been reported [38]. Physical activity is frequently reported to be lower among overweight/obese children [13,29,35,39,40,41]. Studies on associations between SES and children's physical activity have generated inconsistent results. For example, in HICs, children from high socio-economic status (HSES) families were more likely to meet physical activity guidelines [34] whereas in LICs children from low socio-economic status (LSES) were more likely to meet physical activity guidelines [17]. Physical activity is also lower among children from families that own cars [42,43,44,45]. However, there are inconsistent findings on the associations between parental education level and children's physical activity [17,36,42,45]. Therefore, there is still need for more research assessing the sociodemographic correlates of children's physical activity levels, particularly in LICs countries like Uganda where little research has been conducted [11,20]. The present study thus helped to identify children that required immediate intervention

To our knowledge, there is no study that has used accelerometers to measure physical activity levels among school-going children in Kampala city, Uganda. Therefore, the present study assessed the prevalence of accelerometer-measured physical activity intensity levels, compliance with the WHO, 2010 physical activity guidelines and sociodemographic correlates of physical activity among school-going children in Kampala city, Uganda.

## Materials and methods

### Design and participants recruitment

This was a cross-sectional study of a representative sample of school-going children aged 10–12 years old in Kampala city, Uganda. As children aged 10 to 12 years old are transiting from childhood to adolescence, they gain some autonomy in decision making which may be critical

to declines in their physical activity [46,47]. Kampala city is the capital and largest city in Uganda covering an area of 182 km$^2$ with population of 1,516,210 residents from diverse ethnic groups and SES [48]. Kampala comprises of five administrative divisions, that is Nakawa, Makindye, Rubaga, Central and Kawempe [49]. Participants were selected using a multistage random sampling method. In stage one, we randomly selected two out of the five divisions (Central and Nakawa); from which 7 primary schools (3 public schools and 4 private schools) were randomly selected. One classroom from any one grade (5$^{th}$ through 7$^{th}$) was randomly selected and all children from the selected classroom, except those who had physical and health conditions that limited their participation in physical activity were invited to participate in this study. Ethical approval to conduct the study was obtained from the Uganda National Council of Science and Technology (SS4340) and Kenyatta University Ethical Review Board (PKU/ 619/1703). Permission to access schools was granted by the Directorate of Education and Social Services, Kampala Capital City Authority (KCCA). The respective school head teachers, approved the school's participation in the study. A parent/guardian provided written informed consent for themselves and their child in addition to written assent from the child. Data were collected during school sessions from May 2017 through August 2018

## Measures

**Accelerometry.** Children wore a tri-axial ActiGraph GT3X+ (Pensacola, Florida, USA) accelerometer on the right hip using an elastic belt for 7 consecutive days including 2 weekend days. A 24-hour wear protocol was employed to increase compliance [28,50]; and as such children were requested to wear the monitor all the time except when engaging in water-based activities like swimming and bathing. ActiGraph accelerometers are reliable and valid measures of children's physical activity [21,26]. Using Actilife software (version 6.13.3) (ActiGraph, Pensacola, Florida, USA) the fully charged accelerometers were initialized to collect second to second movement counts at midnight following the first day that the children received the accelerometers; at a samplings rate of 80 HZ. Data were downloaded using ActiLife v6.13.3 (ActiGraph, Pensacola, Florida, USA) in raw format as GT3+ files and AGD files with 1 second epoch. The 24-hour protocol required sleep time to be identified and accounted for before evaluating wake wear time and generating physical activity variables of interest [51,52]. We used the Sadeh algorithm, which is in built into the sleep scoring function in Acti-Life software to identify individualised daily sleep on set and offset time for each valid day for each child [53]; this is a valid method for removal of sleep [54]. Daily sleep on set and offset time was used to create time filters in CSV files (Excel Microsoft co-operation, 2016). Time filtered files for the wake period were created and used to identify non wear time and wear time. We defined non-wear as 20 minutes of consecutive 0 counts. Sufficient wear time was determined as 4 days including 1 weekend day with ≥ 10 hours/day. The time spent in different levels of movement intensity were generated basing on the Evenson cut points as: Sedentary time (≤ 25 counts/15 s), LPA (26–573 counts/15 s), MPA (574–1002 counts/15 s) and VPA (≥1003 counts/15 s) [55]. These cut-offs have been recommended as the most accurate for classifying children's physical activity levels [56]. Time spent in MVPA was calculated as the sum of MPA and VPA. Children were classified as meeting the physical activity guidelines (sufficiently active) if their mean amount of time spent in MVPA/day was ≥60 minutes in accordance to the WHO, 2010 physical activity recommendations [4].

**Anthropometry.** Each child had their height (to the nearest 0.1 cm) and body weight (to the nearest 0.1Kg) measured without shoes and with minimal clothing, using a portable stadiometer (Seca 213 portable stadiometer, Hamburg, Germany) and a digital weighing scale (Seca 869 portable electronic digital weighing scale, Hamburg, Germany) respectively following a

standardised procedure. Weight status was calculated as BMI (kilograms per meter squared) and children were categorised as thin/normal weight and overweight/obese using the WHO, 2007 age and gender specific BMI percentiles [57].

**Socio-demographics.** A validated questionnaire assessing children and parents' socio-demographics and neighbourhood built environment [58] was completed by parents/guardians. In this paper, questions that captured children and parents' socio-demographic factors were analysed. Parents reported their children's date of birth (from which the child's actual age at the time of the study was generated) and sex. The questionnaire also captured information about parents' age, sex, marital status, level of education; number of cars at home and the number of children and youth aged 6 to 17years in their homes.

### Recruitment and completion rate

Using the Daniel (1999) formula [59], and an expected prevalence of 21.4% obtained from a previous study by Millstein and colleagues [60] a sample size of 254 was generated. However, because the children were to be sampled in clusters by divisions and schools, the above sample size was multiplied by a design effect of 2 [61] which produced a required sample size of 500 children. To further allow for children who may fail to provide valid and/or incomplete data the enrolment target was set to 600 children. A sample of 600 children received a study package that contained an introduction letter, parent informed consent form, child assent form and a parent/guardian questionnaire to take home to their parents/guardians. Of the 600 children who were invited to participate, 400 (66.7%) had parents/guardians who completed the questionnaire and 328 (54.6%) parental/guardian consented for their children to participate in accelerometry and anthropometric assessment. Of the 328 children who obtained parental consent to wear devices, 256 had valid accelerometry data and were therefore retained for analysis. The response rate was 42.7%. We further assessed demographic characteristics of children who had valid accelerometry results (n = 256) and compared them to those who had complete questionnaire data (n = 400) and found no differences.

### Data analysis

Continuous data such as accelerometer counts were summarised as means and standard deviations while categorical data such as sex were presented as frequencies and percentages. To test for statistical differences between physical activity intensity levels and children's socio-demographic factors, Student's t-tests with unequal variance for factors with two levels and analysis of variance (ANOVA) for factors with more than two levels were used. The two tests were run after testing for assumptions such as equality of variance using the variance ratio test and the Bartlett's test for the t-test and ANOVA respectively. A multi-level mixed effect logistic regression model adjusted for clustering at division and school level was used to examine associations between compliance with physical activity guidelines and each of the socio-demographic variables. We used a backward model fitting technique and set the inclusion into the multivariable model at a $p<0.2$ and also included other factors highlighted in literature such as age and sex. Statistical significance was set at $p<0.05$ and all data were analysed using STATA statistical software Version 14.2.

## Results

### Accelerometer para-data

The para-data presented in Fig 1 was generated during the process of accelerometry enrolment, data collection, management and processing [62]. Of the 400 hundred children who

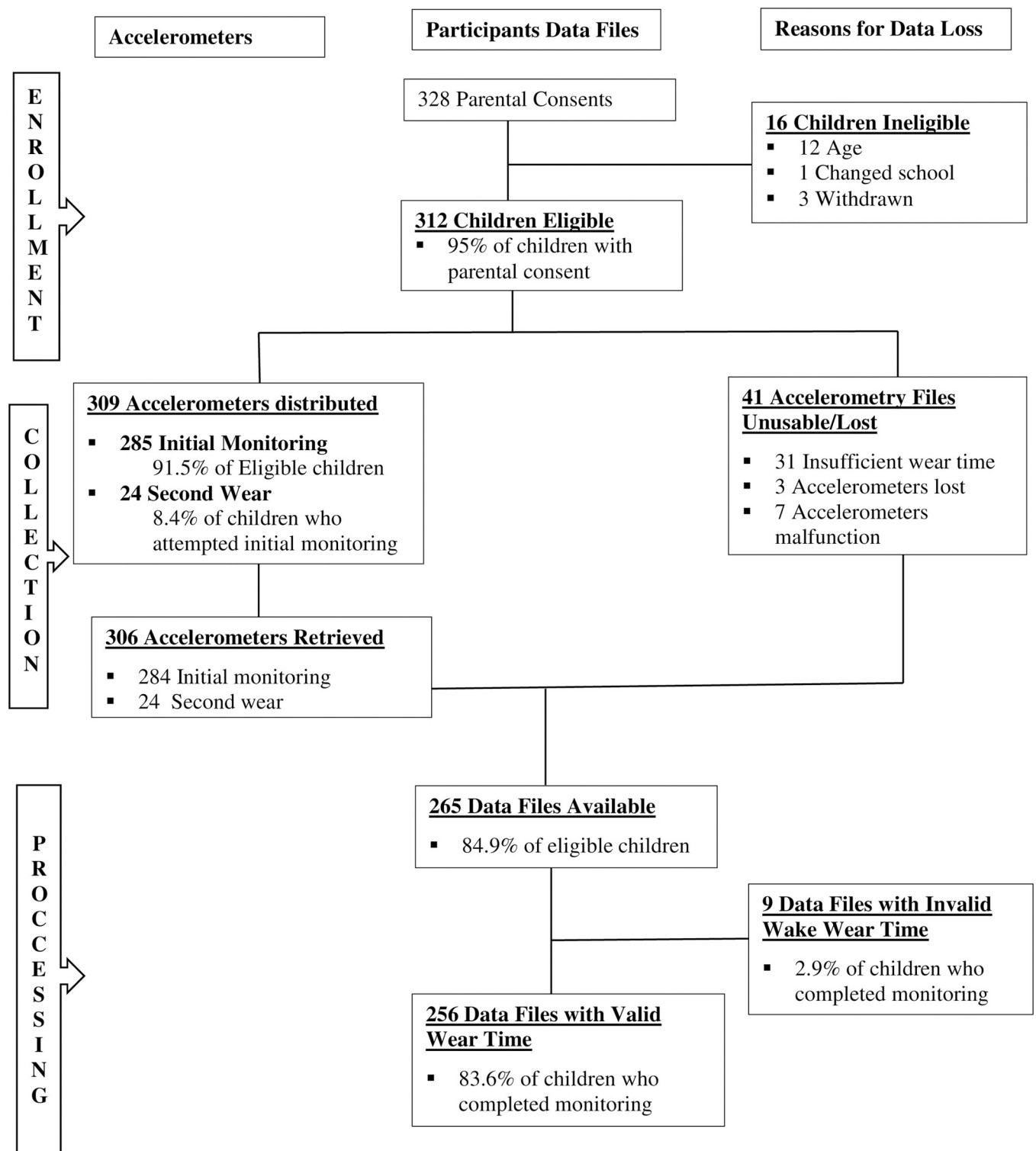

**Fig 1. Participant flow chart reflecting accelerometry stages of participant enrollment, data collection, data processing and reasons for data loss at each stage (adapted from Tudor-Locke et al. 2015).**

returned complete parent/guardian questionnaires, 328 (82%) obtained parent/guardian consent to participate in accelerometry. The children who met the study age criteria were 312. During the entire study, 309 children were monitored; 285 children wore the monitor once whereas 24 children had additional monitoring. After retrieval of monitors and data download 41 data files were invalid mainly due to insufficient wear (31 files), malfunction (7 accelerometers) and loss (3 accelerometers). The final locked data set had 256 files with valid wake wearing time (78% of the children who had parental consent).

## Participant characteristics

Children and parent/guardian characteristics are summarised in Table 1. The study sample comprised 256 children/parent pairs who completed the survey and had valid accelerometer-measured physical activity data. Most of the children attended private schools (58.3%) versus public schools (41.7%). More than half of the children were females (55.9%). Majority of the children were aged 10 and 11 years old (71.5%). Approximately three quarters of the children were of thin/normal weight (79.3%). More than half (58.6%) of parent/guardian respondents were females. Most of the parents/guardians (47.1%) were in the age range of 31 to 40 years old. One in every five parents/guardians were married or living with a partner. Majority of the parents had attained a diploma/degree/postgraduate level of education (74.2%); 70.3% of the families owned a car; and most of the households studied (62.1%) had 2 to 4 children aged 0 to 17 years old. Children wore accelerometers for an average of 15.6 hours/day and 6.5 days and out of the 24 hours and 7 days respectively.

## Physical activity intensity levels by sex, age, type of school and weight status

The children spent most of their time sedentary (9.8±2.1 hours/day), which accounted for 64% of their wake time. They spent another 4.5±0.8 hours /day in LPA and 56±25.7 minutes/day in MVPA with more time accumulated in MPA (38.6±16 minutes/day). Children attending private schools accumulated more sedentary time (P<0.001) compared to their peers from public schools. Children's LPA was significantly different by age (p<0.05). We found significant sex differences in MPA (p<0.05) and VPA (P<0.001), with boys engaging in more MPA and VPA than girls. We also found significant differences in MPA (p<0.001) and VPA (p<0.001) by type of school; children attending public schools accumulated 16.4 and 9.3 more minutes/day of MPA and VPA respectively compared to their peers attending private schools. Thin/normal weight children had significantly higher amounts of MPA (p<0.001) and VPA (P<0.001) compared to overweight/obese children (Table 2).

## Children's compliance with physical activity recommendations

WHO (2010) [4] recommends that children accumulate at least 60 minutes of MVPA daily. Table 3 shows children's compliance with these recommendations by age, sex, type of school and weight status. Only 36.3% of the 256 children participated in ≥ 60 minutes/day of MVPA. Significantly more males (38.9%) than females (34.3%) accumulated recommended MVPA. Significantly more children from public schools (62.3%) than their peers from private schools (18%), met the physical activity guidelines. Significantly more thin/normal weight children (42.9%) engaged in sufficient amounts of physical activity than overweight/obese children (11.3%).

**Table 1. Children and parents/guardians characteristics.**

| Characteristics | Type of school n(%) | | Overall N(%) |
| --- | --- | --- | --- |
| | Private (n = 150) | Public (n = 106) | Overall (N = 256) |
| **Children's characteristics** | | | |
| **Sex** | | | |
| Male | 75 (25.8) | 38 (14.8) | 113 (44.1) |
| Female | 75 (29.3) | 68 (26.6) | 143 (55.9) |
| **Age (years)** | | | |
| 10 | 69 (27.0) | 20 (7.8) | 88 (34.8) |
| 11 | 56 (21.9) | 38 (14.8) | 94 (36.7) |
| 12 | 25 (9.8) | 48 (18.7) | 74 (28.5) |
| **Weight status (Calculated as BMI)** | | | |
| Overweight/Obese | 47 (18.4) | 6 (2.3) | 53 (20.7) |
| Thin/Normal weight | 103 (40.2) | 100 (39.1) | 203 (79.3) |
| **Parents/guardian characteristics** | | | |
| **Sex** | | | |
| Male | 63 (24.6) | 43 (16.8) | 106 (41.4) |
| Female | 87 (34.0) | 63 (24.6) | 150 (58.6) |
| **Marital status** | | | |
| Married/Living with partner | 128 (50.0) | 79 (30.9) | 207 (80.9) |
| Single/Widowed/Divorced | 22 (8.6) | 27 (10.6) | 49 (19.1) |
| **Age** | | | |
| 21–30 | 5 (2.1) | 12 (5.0) | 17 (7.1) |
| 31–40 | 70 (29.4) | 42 (17.6) | 112 (47.1) |
| 41–50 | 55 (23.1) | 42 (17.6) | 97 (40.8) |
| 51–66 | 7 (2.9) | 5 (2.1) | 12 (5.0) |
| **Level of education** | | | |
| Diploma/Degree/Postgraduate | 140 (54.7) | 50 (19.5) | 190 (74.2) |
| Certificate (Ordinary and Advanced level) | 10 (3.91) | 56 (21.9) | 66 (25.8) |
| **Number of cars at home** | | | |
| None | 12 (4.7) | 64 (25.0) | 76 (29.7) |
| One | 65 (25.4) | 31 (12.1) | 96 (37.5) |
| More than one | 73 (28.5) | 11 (4.3) | 84 (32.8) |
| **Number of youths in the Household** | | | |
| 0–1 | 30 (11.7) | 13 (5.1) | 43 (16.8) |
| 2–4 | 97 (37.9) | 62 (24.2) | 159 (62.1) |
| 5+ | 23 (9.0) | 31 (12.1) | 54 (21.1) |
| **Accelerometry wear** | | | |
| *Wear time in hours per day | 16.1 (6.0) | 14.9 (1.3) | 15.6 (4.7) |
| *Wear days | 6.2 (1.0) | 6.3 (1.1) | 6.3 (1.1) |

Data presented as counts and (%) and * means (standard deviation), N = total sample size, n = group sample size, BMI = Body Mass Index.

## Socio-demographics correlates of children's physical activity

Socio-demographic factors associated with meeting physical activity guidelines were presented in Table 4. In the unadjusted model, four of the children and parents' characteristics were significantly associated with meeting physical activity guidelines. Specifically, children were more likely to meet physical activity guidelines if they attended a public school (OR = 7.5; 95% CI = 4.24–13.32), were thin/normal weight (OR = 5.88; 95% CI = 2.30–15.00); or if their

**Table 2. Average daily minutes of physical activity at various intensity levels by age, sex, type of school and weight status.**

| | Physical Activity intensity levels (Mean [SD]) | | | |
|---|---|---|---|---|
| | Sedentary time | P-value | LPA | P-value |
| **Overall** | 590.6 (124.0) | | 273 (48.3) | |
| **Sex[#]** | | | | |
| Male | 606.1 (146.6) | 0.089 | 272.5 (48.0) | 0.783 |
| Female | 578.4 (101.6) | | 274 (48.8) | |
| **Age[^]** | | | | |
| 10 | 591.0 (112.0) | 0.995 | 283.3 (44.2) | 0.020 |
| 11 | 591.2 (142.6) | | 272.9 (50.4) | |
| 12 | 589.4 (113.5) | | 262.1 (48.6) | |
| **Type of School[#]** | | | | |
| Private | 617.4 (142.1) | <0.001* | 277.1 (45.4) | 0.155 |
| Public | 552.6 (78.7) | | 268.3 (52.1) | |
| **Weight status[#]** | | | | |
| Overweight/obese | 614.4 (144.2) | 0.117 | 275.3 (47.6) | 0.758 |
| Thin/normal weight | 584.4 (117.8) | | 272.9 (48.6) | |
| | MPA | P-value | VPA | P-value |
| Overall | 38.6 (16.0) | | 17.3 (12.3) | |
| **Sex[#]** | | | | |
| Male | 39.4 (16.3) | 0.002 | 20.7 (15.2) | <0.001* |
| Female | 37.9 (15.8) | | 14.9 (8.7) | |
| **Age[^]** | | | | |
| 10 | 36.9 (13.9) | 0.459 | 15.9 (8.2) | 0.090 |
| 11 | 39.4 (17.9) | | 17.0 (11.0) | |
| 12 | 39.6 (16.0) | | 20.0 (17.1) | |
| **Type of School[#]** | | | | |
| Private | 31.8 (12.1) | <0.001* | 13.6 (7.0) | <0.001* |
| Public | 48.2 (16.0) | | 22.9 (15.8) | |
| **Weight status[#]** | | | | |
| Overweight/Obese | 30.0 (10.9) | <0.001 | 11.6 (6.6) | <0.001 |
| Thin/normal weight | 40.8 (16.4) | | 19.0 (13.0) | |

SES = socio-economic status, HSES = High socio-economic status, LSES = Low socio-economic status. Analysis: mean difference,

T-Test[#],

One-way ANOVA[^],

*p<0.001

mothers reported a lower level of education (OR = 3.64; 95% CI = 2.12–6.24). Lower odds of meeting guidelines were noted for children from families that owned a car (OR = 0.23; 95% CI = 0.14–0.38). In the fully adjusted model, the observed associations of weight status, maternal level of education and car ownership remained significant and their effect size remained nearly unchanged. Specifically, thin/normal weight children (OR = 4.08; 95% CI = 1.42–11.76) and children whose mothers reported lower levels of education (OR = 2.43; 95% CI = 1.84–3.21) were more likely to meet the physical activity guidelines. However, lower odds of meeting physical activity guidelines were noted in children aged 12 years (OR = 0.68; 95% CI = 0.55–0.84) and those from families that owned a car (OR = 0.31; 95% CI = 0.17–0.55). Sex was not significantly associated with meeting physical activity guidelines.

**Table 3. Compliance with physical activity guidelines by children's sociodemographics.**

| Characteristic | Sufficient PA | Insufficient PA | P-value |
|---|---|---|---|
| **Overall** | **n (%)** | **n (%)** | |
| | **93 (36.3%)** | **163 (63.7%)** | |
| **Sex** | | | |
| Male n = 113 | 44 (38.9) | 69 (61.1) | <0.001* |
| Female n = 143 | 49 (34.3) | 94 (65.7) | |
| **Age** | | | |
| 10 years n = 88 | 29 (32.6) | 60 (67.4) | |
| 11 years n = 94 | 35 (37.2) | 59 (62.8) | 0.064 |
| 12 years n = 74 | 29 (39.7) | 44 (60.3) | |
| **School Type** | | | |
| Private n = 150 | 27 (18) | 123 (82) | <0.001* |
| Public n = 106 | 66 (62.3) | 40 (37.7) | |
| **Body Weight Status** | | | |
| Overweight/obese | 6 (11.3) | 47 (88.7) | <0.001* |
| Thin/normal weight | 87 (42.9) | 116 (57.1) | |

PA = physical activity, n = subtotal,

* p<0.001.

## Discussion

The current study assessed accelerometer-measured physical activity intensity levels, compliance with physical activity guidelines and socio-demographic correlates of meeting physical activity guidelines among 10 to 12 years old school-going children in Kampala, city Uganda. The results showed that children spent most of their time sedentary (64%) and in LPA and less time in MVPA. Only 36.3% met the physical activity guidelines, with the proportion of meeting physical activity guidelines lower among girls, private school and overweight/obese children. The adjusted model showed that thin/normal weight children and children whose mothers reported a lower level of education were greater than twice as likely to meet physical activity guidelines; whereas older children and children from families that owned a car had lower odds of meeting physical activity guidelines.

In line with our results, literature shows that a typical physical activity pattern for children comprise of >40% sedentary time [63,64], a substantial amount of time in LPA [3,16,38,63] and <5% of wake time in MVPA [2]. For example, in a review study, Elmesmari et al. reported that children spent >70% of their wake time in sedentary pursuits [65]. In Dakar Senegal, Diouf et al. reported 65% sedentary time among school children 8 to 11 years old [16]. Among Kenyan children, Ojiambo and colleagues found that 72% of children's wake time was sedentary time [14]; whereas Muthuri et al. reported 6.6 hours of sedentary time [17]. This is worrying because sedentary time plays a major role on poor health and overall mortality independent of participation in physical activity [66,67]. Also, sedentary time competes for time children spend in physical activity which may hinder them from achieving the set physical activity guidelines [68,69]. Sedentary time was particularly high among overweight/obese children. Likewise, in a systematic review, Elmesmari et al., found that sedentary time was significantly higher in obese than non-obese groups [65].

The consistent finding that LPA contributes a substantial amount to children's physical activity is supported by findings of the current study [3,17,38,39]. LPA is linked to cardio-metabolic health in children and may be an easier substitute for sedentary time due to its light

**Table 4. Multi-level logistic regression results for associations between children's socio-demographics and compliance to physical activity guidelines.**

| Characteristics | Physical Activity Guidelines n (%) | | Crude OR (95% CI) | P-value | Adjusted OR (95%CI) | P-Value |
|---|---|---|---|---|---|---|
| | Sufficient PA | Insufficient PA | | | | |
| **Type of School | | | | | | |
| Private | 27 (18.0) | 123 (82.0) | 1.00 | | | |
| Public | 66 (62.3) | 40 (37.7) | 7.52(4.24,13.32) | <0.001* | | |
| **Sex** | | | | | | |
| Female | 49 (34.3) | 94 (65.7) | 1.00 | | 1.00 | |
| Male | 44 (38.9) | 69 (61.1) | 1.22 (0.71,2.11) | 0.469 | 1.7 (0.98,2.97) | 0.061 |
| **Age (years)** | | | | | | |
| 10 | 29 (32.6) | 60 (67.4) | 1.00 | | 1.00 | |
| 11 | 35 (37.2) | 59 (62.8) | 1.23 (0.86,1.74) | 0.252 | 0.86 (0.43,1.72) | 0.661 |
| 12 | 29 (39.7) | 44 (60.3) | 1.36 (0.55,3.41) | 0.507 | 0.68 (0.55, 0.84) | <0.001* |
| **Weight status** | | | | | | |
| Overweight/obese | 6 (11.3) | 47 (88.7) | 1.00 | | 1.00 | |
| Normal weight | 87 (42.9) | 116 (57.1) | 5.88 (2.30, 15.00) | <0.001* | 4.08 (1.42,11.76) | 0.009 |
| **Marital status** | | | | | | |
| Married/Living with partner | 70 (33.8) | 137 (66.2) | 1.00 | | 1.00 | |
| Single/Widowed/Divorced | 23 (46.9) | 26 (53.1) | 1.73 (0.79,3.77) | 0.167 | 1.14 (0.54,2.41) | 0.732 |
| **Mother's education level** | | | | | | |
| Diploma/Degree/Postgraduate | 54 (28.4) | 136 (71.6) | 1.00 | | 1.00 | |
| Certificate (Ordinary and Advanced level | 39 (59.1) | 27 (40.9) | 3.64 (2.12, 6.24) | <0.001* | 2.43 (1.84,3.21) | <0.001* |
| **Number of cars at home** | | | | | | |
| None | 47 (61.8) | 29 (38.2) | 1.00 | | 1.00 | |
| One | 26 (27.1) | 70 (72.9) | 0.23 (0.14,0.38) | <0.001* | 0.30 (0.22,0.40) | <0.001* |
| More than one | 20 (23.8) | 64 (76.2) | 0.19 (0.05,0.76) | 0.019 | 0.31 (0.17,0.55) | <0.001* |
| **Children and youths(6 to 17 years) in the Household** | | | | | | |
| 0–1 | 15 (34.9) | 28 (65.1) | 1.00 | | | |
| 2–4 | 54 (34.0) | 105 (66.0) | 0.96 (0.48,1.90) | 0.907 | | |
| 5+ | 24 (44.4) | 30 (55.6) | 1.49 (0.53,4.21) | 0.448 | | |

n = subtotal, ** clustered at school level,

* p<0.001.

intensity [2,39]. However, higher intensity physical activity (MPA & VPA) is linked to greater health benefits [1,2,70], particularly VPA which is favourable for obesity prevention [2,38,39,40]. However, similar to literature, our results showed that children spent less time in MVPA, the highest percentage coming from MPA [16,17,38,39]. Although children may not be able to sustain high intensity physical activity for a long period of time, shorter bouts of VPA may have greater health benefits than longer bouts of MPA [3,70]. Therefore, interventions programs focusing on increasing physical activity levels (MVPA) and decrease sedentary time are needed.

The average time spent in MVPA among school-going children in Kampala was 56 minutes/day which was less than the recommended minimum of 60 minutes/day. Only 36.3% of the children met the WHO, 2010 physical activity guidelines. Literature also shows that children do not engage in sufficient amounts of MVPA [9,10,11,16,17,29,30,41]. For example, results from ISCOLE, Kenya, showed that children aged 9 to 11 years recorded an average of 36 minutes/day of MVPA, and only 12.6% of the children met the physical activity guidelines [17]. Differences in MVPA by children's characteristics revealed that girls, private school and

overweight/obese children were less likely to meet the physical activity guidelines. Sex differences in children's MVPA favoring boys have been consistently reported in literature [11,16,17,29,34,35,41,71] and the present study confirms these findings. Cultural factors may explain the sex differences in children's MVPA [2]. Culture determines the roles taken on by boys and girls which influences their physical activity behaviour and interests [72,73]. Furthermore, boys have higher independent mobility which provides them with more opportunities to engage in physical activity [74]. Similar to results from the ISCOLE study conducted in Nairobi Kenya, a higher percentage of children in public schools accumulated more MVPA compered to their peers in private schools [17]. The results of our study were in line with those from previous studies that a higher proportion of thin/normal weight children meet physical activity guidelines compared to their overweight/obese peers [35,39,65].

Physical activity was inversely correlated with children's weight status; specifically, overweight/obese children were unlikely to meet physical activity guidelines. This finding is consistent with literature [17,29,35,39,65]. Nevertheless, among urban and rural children aged 11 to 16 years in Uganda, high weight status was associated with sufficient physical activity; however, the highest weight status identified in this study was normal weight [33]. Inconsistent associations between physical activity and weight status have also been reported [75] whereas some studies found no significant associations [71]. The inconsistent findings may be due to the different criteria used to define weight status (WHO, US Centre for Disease Control and Prevention [CDC] and International Obesity Task Force [IOTF]) which give different estimates [65]. Our findings should also be viewed with caution due to a possibility of reverse causation.

The observation that older children were less likely to meet physical activity guidelines is consistent with previous studies demonstrating that children's physical activity declines with increasing age [18,29,70]. LSES (as indicated by low maternal level of education and no family car) was positively associated with meeting physical activity guidelines. A review of studies from Sub-Saharan Africa [15] other studies [13–17] reported similar results. On the contrary, studies from HICs [34,36,41,43] reported positive associations between children's physical activity and HSES. The contradictory results may be explained by the different proxy indicators used to assess SES [76]. In addition, in LICs like Uganda it may be a necessity rather than a choice for children from LSES families to engage in physical activity; whereas for children from HSES families, technological advances like car ownership may hinder their participation in physical activity, and for them to be active may require a more deliberate initiative [77]. The Negative association between higher levels of maternal education and children's physical activity found in this study have been reported elsewhere [17,38,45,78]. Crawford and colleagues proposed that highly educated mothers may not have time to model physical activity behaviour for their children because of full time employment [79]. Results of the current study also showed an inverse association between owning a car and meeting physical activity guidelines. Similar findings have been reported elsewhere [43,45]. Owning one or more cars is a disincentive to active travel which is a major contributor to children's physical activity [17,43,45,71,80].

Therefore, there is need for developing effective strategies and policies with the aim of increasing physical activity levels among school going children in Kampala city and Uganda. This may be achieved by implementing strategies and policies that have been proposed by various global and regional organisations including those of the Active Healthy Kids Global Alliance (AHKGA) in the fight against the insufficient physical activity among children [81,82]. The current study further highlighted the need for nationally representative physical activity data. The Ministry of Education and Sports in Uganda should fund the development and release of a national report card on physical activity for children in Uganda for surveillance and promotion of physical activity among Ugandan children.

A particular strength of this study was the use of accelerometers to measure children's physical activity which provided a more robust assessment than self-report measures. This is also the first study of this kind to be conducted in Uganda. However, when using accelerometers there are some limitations in quantifying physical activity of children who engage in swimming, cycling, and activities that predominantly involve upper body movements and weight lifting [83,84]; therefore, we may have underestimated children's physical activity. However, according to the education abstract, 2014 children in Uganda rarely engage in cycling and swimming [83]. We also used the more liberal criteria in which participating in an average of ≥60 minutes of MVPA on all measured days was considered sufficient physical activity. It is likely that some of the children were not meeting the ≥ 60 minutes of MVPA on all 7days of the week as stated in the guidelines [4]. The study is also not nationally representative; therefore, the results cannot be generalized to all school-going children in Ugandan. The current study findings should be interpreted with caution given the cross sectional design which makes it impossible to infer causality and the low response rate

## Conclusion

In conclusion the current study findings revealed that children spend substantial time in sedentary pursuits and LPA and less time in MVPA. Most of the children in did not meet the physical activity guidelines of ≥60 minutes of MVPA every day. MVPA was higher among boys, public school and thin/normal weight children. Specific interventions are needed to help children in Kampala city to increase their physical activity levels; particularly girls, overweight/obese children, and children from families that have highly educated parents and own cars. Although the response rate was relatively low, this study may be important for surveillance and serve as a model for a nationwide study.

## Supporting information

**S1 Appendix. Data set.**
(XLSX)

## Acknowledgments

The authors appreciate the Directorate of Education and Social Services, Kampala Capital City Authority (KCCA) for permitting them to access schools. We appreciate the research assistants who greatly contributed to data collection. We are also grateful to all school head teachers, teachers, parents/guardians and children who participated in this study. We thank the Physical Activity and Health Laboratory at the University of Massachusetts Amherst, USA for support on accelerometry data management and interpretation.

## Author Contributions

**Conceptualization:** Bernadette Nakabazzi, Lucy-Joy M. Wachira, Adewale L. Oyeyemi, Vincent O. Onywera.

**Formal analysis:** Ronald Ssenyonga.

**Funding acquisition:** Bernadette Nakabazzi.

**Investigation:** Bernadette Nakabazzi.

**Methodology:** Bernadette Nakabazzi, Lucy-Joy M. Wachira, Adewale L. Oyeyemi, Vincent O. Onywera.

**Supervision:** Lucy-Joy M. Wachira, Adewale L. Oyeyemi, Vincent O. Onywera.

**Visualization:** Bernadette Nakabazzi, Ronald Ssenyonga.

**Writing – original draft:** Bernadette Nakabazzi.

**Writing – review & editing:** Bernadette Nakabazzi, Lucy-Joy M. Wachira, Adewale L. Oyeyemi, Ronald Ssenyonga, Vincent O. Onywera.

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
