## [Decision Letter · Decision Letter 0]

24 Feb 2020

PONE-D-20-00496

Prevalence and sociodemographic correlates of accelerometer measured physical activity levels of school-going children in Kampala city, Uganda.

PLOS ONE

Dear Ms Nakabazzi,

Thank you for submitting your manuscript to PLOS ONE. After careful consideration, we feel that it has merit but does not fully meet PLOS ONE’s publication criteria as it currently stands. Therefore, we invite you to submit a revised version of the manuscript that addresses the points raised during the review process.

The current manuscript describes PA patterns among children from Uganda. There is a lack of information of this type in countries from Africa, so it is to important to create evidence in the scientific literature. However, our reviewers consider that the document should be improved deeply. In this sense, I strongly recommend you to meet all the suggestions they provide throughout the revision letters in order to have a manuscript that could be published.

We would appreciate receiving your revised manuscript by Apr 09 2020 11:59PM. To enhance the reproducibility of your results, we recommend that if applicable you deposit your laboratory protocols in protocols.io, where a protocol can be assigned its own identifier (DOI) such that it can be cited independently in the future. For instructions see: http://journals.plos.org/plosone/s/submission-guidelines#loc-laboratory-protocols

We look forward to receiving your revised manuscript.

Kind regards,

Javier Brazo-Sayavera, Ph.D.

Academic Editor

PLOS ONE

Journal Requirements:

Please ensure that your manuscript meets PLOS ONE's style requirements, including those for file naming. The PLOS ONE style templates can be found at http://www.plosone.org/attachments/PLOSOne_formatting_sample_main_body.pdf and http://www.plosone.org/attachments/PLOSOne_formatting_sample_title_authors_affiliations.pdf

Reviewers' comments:

Reviewer's Responses to Questions

**Comments to the Author**

1. Is the manuscript technically sound, and do the data support the conclusions?

Reviewer #1: No

Reviewer #2: Partly

2. Has the statistical analysis been performed appropriately and rigorously? 

Reviewer #1: No

Reviewer #2: I Don't Know

3. Have the authors made all data underlying the findings in their manuscript fully available?

Reviewer #1: Yes

Reviewer #2: Yes

4. Is the manuscript presented in an intelligible fashion and written in standard English?

Reviewer #1: Yes

Reviewer #2: Yes

5. Review Comments to the Author

Reviewer #1: Abstract:

----------

* Line 23 (and throughout the manuscript): Could the authors please use the term "device-based" instead of "objective"? The so-called "objective measurement" of physical activity includes several subjective calls to make by the researchers (which device to use, which analysis algorithm to use, input acceleration signal, cut-offs, etc.).

*Results (lines 36/37): For sedentary behaviour and MVPA, were data normally distributed? What were the standard deviations for the estimates?

*Results (lines 38/40): Were probability ratios measured or odds ratio? From the 95% CIs (0.71 - 2.11), looks like this estimate is not statistically significant.

*Results (lines 39/40): Please define a term before using the abbreviated form. LSES = low SES? HSES = high-SES?

Conclusion (line 45): It seems like the authors are assuming that sedentary time would be displaced by physical activity, which can happen. However, as this paper is about the prevalence and socio-demographic correlates of physical activity, it may not be ideal for making the assumption here in the Abstract!

Introduction:

---------------

* Line 63 (ref 9): The authors could consider citing a more recent paper about the prevalence of insufficient physical activity among children and adolescents: Guthold et al 2020 Global trends in insufficient physical activity among adolescents: a pooled analysis of 298 population-based surveys with 1·6 million participants. The Lancet Child & Adolescent Health 4:23-35. This study has data from 16 sub-Saharan African countries. The authors should consider re-framing their study background using information from this study.

* Line 68 (ref 12): This is a debated area. The authors should use "stronger" evidence - meta-analysis, systematic review - to support their statement.

* Line 98: The term "subjective measure" should be "recall-based measure" or "questionnaire-based measure".

* Line 106: The authors should acknowledge that "overweight and obesity" is not an SES measure.

Materials and methods:

* Design and Participants Recruitment subsection (line starting on 119): The authors should justify studying this age group. The recommendation for 60 min/d physical activity is for 5-17-year-old children and adolescents. Why did the authors study only children those aged 10-12 years?

* Design and Participants Recruitment subsection (line 127): So, everyone in those classrooms was eligible to participate. Were there any exclusion criteria? Did the authors ask any question about health conditions that, potentially, restrict some students' physical activity participation?

* Design and Participants Recruitment subsection (line 133): Could the authors please clarify what "during school sessions" mean? Were data not collected during holidays?

* Participants' Sociodemographics subsection (about SES measure; lines 168/170): I am not aware of how public-private schools operate in Kampala, and how they are viewed concerning SES. There might be several reasons for studying in a public school. I am unsure if this is an appropriate measure of SES -- this is an unusual approach to me. The authors may consider labelling this variable as "Type of school" not as a proxy for SES.

* Recruitment and Completion rate subsection: How did the authors calculate the sample size for this study?

* Data Analysis subsection (lines 182/183): Did the authors check if the data were normally distributed. If data were not normally disturbed, mean and SD would not be the appropriate summary statistics, the median and interquartile range would be.

* Data Analysis subsection (line 186): Could the authors please confirm all relevant assumptions for ANOVA were met?

* Data Analysis subsection (line 188): Did the regression model adjust for the nested nature of the data (i.e., a multilevel model)? Was this at the level of the division or also at the level of the school, or classroom? As the students are nested at different levels during their selection process, how the analyses account for this clustering?

* Data Analysis subsection (lines 188/191): Did the authors include all these variables in the model? Did the authors check for collinearity? It is likely that SES (i.e., type of school), maternal education, and car ownership would have multi-collinearity as these perhaps measure the same (or a similar) construct. Including all these three variables in the same model may make the model unstable and may not give correct estimates. How did the authors build their model? Did the authors run any post-estimation diagnostic test for their model?

* Data Analysis subsection (line 190): Weight status is not a socio-demographic variable.

Results:

---------

* Table 1: Is there any reason to describe the participants by their type of school?

* Line 249: I think the right way to express this is "Significantly more males (38.9%) than females (34.3%) accumulated recommended MVPA". The same applies to the rest of the paragraph.

* Table 4 (in general): I wonder why the authors presented crude ORs.

* Table 4 (variable sex): From the 95% CIs for "sex" (0.71 - 2.11) it seems like the p-value should not be 0.046.

* Table 4 (variable weight status): Some numbers in the cells are too small to run a logistic regression analysis. The CIs for the obese group is very wide and perhaps suggest that the model was unstable. I am not sure if combining the overweight and obese group into one would help.

Discussion (in general):

The methodological aspects of the paper needs to be revisited before discussing the findings.

Reviewer #2: Please refer to the attached reviewer comments, suggestions and questions. I think this is an important manuscript but it needs some important revisions prior to publication in my opinion. I am not sure the main statistical approach of using bivariate logistic regression was the best given the hierarchical (Individual, classroom, school) nature of these data. Although school environment is included in the analysis, the classroom is not. In light of the foregoing, I hesitate to render final judgement as to the statistical rigor of the analyses. I suggested a number of corrections hence my selecting partly on whether the manuscript is technically sound.

6. PLOS authors have the option to publish the peer review history of their article (what does this mean?). If published, this will include your full peer review and any attached files.

Reviewer #1: No

Reviewer #2: No

---

## [Author Response · Author response to Decision Letter 0]

22 Apr 2020

We are grateful to the editor and the two reviewers for their numerous and detailed comments on our manuscript.

Please find below our responses to each point raised by the academic editor and reviewers. In particular, the methods section has been expanded and made clear also we have re-analysed the data and made relevant changes to the discussion.

We believe that we have addressed all the concerns and our manuscript has greatly improved for publication. 

Sincerely,

On behalf of all the authors,

Bernadette Nakabazzi 

Academic Editor:

No comments provided 

Reviewer #1

Abstract 

1. Line 23 (and throughout the manuscript): Could the authors please use the term "device-based" instead of "objective"? The so-called "objective measurement" of physical activity includes several subjective calls to make by the researchers (which device to use, which analysis algorithm to use, input acceleration signal, cut-offs, etc.).

As requested we have changed the term “objective” and instead used “accelerometer measured” on line 23 and throughout the manuscript in the revised version. We did this to further specify the devise used to measure physical activity in the study.

2. Results (lines 36/37): For sedentary behaviour and MVPA, were data normally distributed? What were the standard deviations for the estimates?

Yes, the data were normally distributed evidence shown below (Comment 6 in the materials and methods section). We have also included the standard deviations for the estimates in the revised version line 36/37 to read; Children’s sedentary time was 9.8±2.1 hours/day and MVPA was 56±25.7 minutes/day.

3. Results (lines 38/40): Were probability ratios measured or odds ratio? From the 95% CIs (0.71 - 2.11), looks like this estimate is not statistically significant. 

We measured odds ratios and this has been corrected in the revised version line 40 to 43.

Yes, the estimate was not statistically significant. The p-value is 0.469 it was a typing error. We have corrected this error. 

4. Results (lines 39/40): Please define a term before using the abbreviated form. LSES = low SES? HSES = high-SES?

We thank the reviewer for pointing this out, we have corrected it (line 41/43) in the revised version.

5. Conclusion (line 45): It seems like the authors are assuming that sedentary time would be displaced by physical activity, which can happen. However, as this paper is about the prevalence and socio-demographic correlates of physical activity, it may not be ideal for making the assumption here in the Abstract!

We agree with this and have made necessary corrections, because the correlates of sedentary time were not assessed in the current study (line 54 revised version)

Introduction

6. Line 63 (ref 9): The authors could consider citing a more recent paper about the prevalence of insufficient physical activity among children and adolescents: Guthold et al 2020 Global trends in insufficient physical activity among adolescents: a pooled analysis of 298 population-based surveys with 1·6 million participants. The Lancet Child & Adolescent Health 4:23-35. This study has data from 16 sub-Saharan African countries. The authors should consider re-framing their study background using information from this study.

We thank the reviewer for suggesting this very important recent paper, we have included the information particularly the global statistics, those for Sub Saharan Africa (SSA) and the gender differences. We noted that the study generated most findings from self-reported data, which covered only 36% of the population in SSA and the trend of data were skewed to high income countries. Also the Global School-based Students Health Survey (GSHS) carried out in Uganda which informed this study focused on adolescents 13 to 17 years old. 

However, we have re-phrased the study background as requested (Line 73/75 and 86/87 in revised version).

7. Line 68 (ref 12): This is a debated area. The authors should use "stronger" evidence - meta-analysis, systematic review - to support their statement.

We have deleted this statement, given that the study is focusing on physical activity and not sedentary time as noted in reviewer comment 5 above. We could also not identify a systematic review or meta-analysis to support the statement.

8. Line 98: The term "subjective measure" should be "recall-based measure" or "questionnaire-based measure".

Line 116/117 in the revised version; The term “subjective measure has been replaced with “recall-based measures” 

9. Line 106: The authors should acknowledge that "overweight and obesity" is not an SES measure.

Children’s physical activity is consistently associated with various sociodemographic factors [29,34,35,36]. Various studies that used both objective and recall-based measures of physical activity consistently reported sex differences in physical activity favouring boys [11,17,29,34,35,36,37]. Children’s physical activity has also been found to decline with increasing age [18,29], nevertheless non-significant associations have also been reported [38]. Physical activity is frequently reported to be lower among overweight and obese children [13, 29,35,39,40,41]. Studies on associations between SES and children’s physical activity have generated inconsistent results. For example, in HICs, children from HSES families were more likely to meet physical activity guidelines [34] whereas in SSA children from HSES were unlikely to meet physical activity guidelines [17]. Physical activity is also lower among children from families that own cars [42,43,44,45]. However, there are inconsistent findings on the associations between parental education level and children’s physical activity [17, 36, 42,45]. Therefore, there is still need for more research assessing the sociodemographic correlates of children’s physical activity levels, particularly in SSA countries like Uganda where little research has been conducted [11,20]. The present study thus helped to identify children that required immediate intervention

Materials and methods:

1. Design and Participants Recruitment subsection (line starting on 119): The authors should justify studying this age group. The recommendation for 60 min/d physical activity is for 5-17-year-old children and adolescents. Why did the authors study only children those aged 10-12 years?

We thank the reviewer for pointing out this, we have made a justification for studying this age group in line 145/147 in the revised version to read as; 

As children aged 10 to 12 years old are transiting from childhood to adolescence, they gain some autonomy in decision making which is critical to declines in their physical activity [43,44]. 

Younger participants have not gained independence in choosing and guiding their activities and behaviour and are still greatly influenced (restricted by parents and guardians for several reasons). Older children are greatly influenced by the pubertal growth spurt which could have influenced several aspects also tested in our study such as weight and adiposity status

2. Design and Participants Recruitment subsection (line 127): So, everyone in those classrooms was eligible to participate. Were there any exclusion criteria? Did the authors ask any question about health conditions that, potentially, restrict some students' physical activity participation?

The classroom approach was meant to be as inclusive as possible so that learners do not feel left out apart from those that had conditions likely to interfere with physical activity at the time of study. 

We excluded children who had physical and health conditions that limited their participation in physical activity (Line 154/155 in the revised edition)

3. Design and Participants Recruitment subsection (line 133): Could the authors please clarify what "during school sessions" mean? Were data not collected during holidays?

During school session means the data were collected when children were in school and not during holidays. 

The objective was to assess the children’s’ typical physical activity behaviour and since children spend most of their time in school the study also focused on school season (i.e. activity before school starts, during school program, after school ends and weekends) to later advise school based interventions.

4. Participants' Sociodemographics subsection (about SES measure; lines 168/170): I am not aware of how public-private schools operate in Kampala, and how they are viewed concerning SES. There might be several reasons for studying in a public school. I am unsure if this is an appropriate measure of SES -- this is an unusual approach to me. The authors may consider labelling this variable as "Type of school" not as a proxy for SES.

We thank the reviewer for pointing this out. We have labelled this variable as type of school. However, we have used maternal level of education and car ownership as indicators of SES; that is high maternal level of education (diploma/degree/postgraduate) and households with one or more cars to represent HSES and low maternal level of education (Ordinary level/advanced level) and households with no cars for LSES.

5. Recruitment and Completion rate subsection: How did the authors calculate the sample size for this study?

We used the Daniel (1999) formula to generate a sample size of 254. However, because the children were to be sampled in clusters by divisions and schools, the above sample size was multiplied by a design effect of 2 (Conchran,1977) which produced a required sample size of 500 children. To further allow for children who may fail to provide valid and/or incomplete data the enrolment target was set to 600 children (line 209/2140) in the revised version

6. Data Analysis subsection (lines 182/183): Did the authors check if the data were normally distributed. If data were not normally disturbed, mean and SD would not be the appropriate summary statistics, the median and interquartile range would be.

Yes, the authors check for normal distribution using two methods the graphical and statistical. Both are presented below, the graph points normal distribution and the Schapiro Wilk test gave a p-value of 0.140 both showing normal distribution.

7. Data Analysis subsection (line 186): Could the authors please confirm all relevant assumptions for ANOVA were met?

Yes, the all relevant assumptions where tested for and met.

1. Normality- we used the raw data and not the normality of the errors and arrived at the same result. The example of age is shown in the graph below

2. Independence, 

Data were independent.

3. Homoscedasticity, Using the Bartlett’s test as shown below for the age variable.

8. Data Analysis subsection (line 188): Did the regression model adjust for the nested nature of the data (i.e., a multilevel model)? Was this at the level of the division or also at the level of the school, or classroom? As the students are nested at different levels during their selection process, how the analyses account for this clustering?

Yes, the regression model did adjust for the nested nature of the data. We used a Multilevel mixed-effects logistic regression where we adjusted for both division and school (Line 231/236 in revised edition), as shown below. 

melogit outcome exposure || division || school:, or

9. Data Analysis subsection (lines 188/191): Did the authors include all these variables in the model? Did the authors check for collinearity? It is likely that SES (i.e., type of school), maternal education, and car ownership would have multi-collinearity as these perhaps measure the same (or a similar) construct. Including all these three variables in the same model may make the model unstable and may not give correct estimates. How did the authors build their model? Did the authors run any post-estimation diagnostic test for their model?

Yes, we checked for collinearity using the Variance Inflation Factor (VIF)s by variable as well as the overall VIF value. All these were not much greater than one suggesting no collinearity among the variables modelled.

We used the backward model building technique which allowed all variables to be evaluated at the start thus minimising negative confounding and ruling out collinearity as well as instability of the model. We also used the Akaike's and Schwarz's Bayesian information criteria (AIC and BIC) as post estimation to assess whether the final model was better than those previously fitted on our data.

10. Data Analysis subsection (line 190): Weight status is not a socio-demographic variable.

Yes, we agree with the reviewer, however weight status in the present study was used as a primary characteristic of children.

Results:

1. Table 1: Is there any reason to describe the participants by their type of school?

Yes, because at the design of the study, there was evidence of disparities between the two types of schools especially in terms of commuting to and from school; which directly contributed to the outcome that we sought to measure that is physical activity. Therefore, this description provides an assessment on whether there could have been any marked differences in the type of school across the factors studied such as weight status where most overweight and obese children were from private schools. This provides context for interpretation of our results.

2. Line 249: I think the right way to express this is "Significantly more males (38.9%) than females (34.3%) accumulated recommended MVPA". The same applies to the rest of the paragraph.

We thank the reviewer for pointing out this, we have revised the paragraph to read; Significantly more males (38.9%) than females (34.3%) accumulated recommended MVPA. Significantly more children from public schools (62.3%) than their peers from private schools (18%), met the physical activity guidelines. Significantly more thin/normal weight children (42.9%) engaged in sufficient amounts of physical activity than overweight/obese children (11.3%). (Line 299/300 and 306/309 revised edition)

3. Table 4 (in general): I wonder why the authors presented crude ORs.

We have corrected this and presented both crude and adjusted OR (Table 4 revised edition)

4. Table 4 (variable sex): From the 95% CIs for "sex" (0.71 - 2.11) it seems like the p-value should not be 0.046. 

Yes, this p-value was not 0.046, this was a typing error. We have corrected this and the p-value was 0.469.

5. Table 4 (variable weight status): Some numbers in the cells are too small to run a logistic regression analysis. The CIs for the obese group is very wide and perhaps suggest that the model was unstable. I am not sure if combining the overweight and obese group into one would help.

We thank the reviewer for this suggestion, yes some numbers are too small, we have combined the thin and normal weight group and the overweight and obese groups. We also found a similar problem with the maternal level of education and combined ordinary level and advanced level, and diploma, degree and postgraduate.

Discussion (in general):

The methodological aspects of the paper needs to be revisited before discussing the findings.

We have made the necessary revisions in the methodology as requested and re-revised the discussion as presented in the revised version (Line 346/432).

Reviewer #2:

Please refer to the attached reviewer comments, suggestions and questions. I think this is an important manuscript but it needs some important revisions prior to publication in my opinion. I am not sure the main statistical approach of using bivariate logistic regression was the best given the hierarchical (Individual, classroom, school) nature of these data. Although school environment is included in the analysis, the classroom is not. In light of the foregoing, I hesitate to render final judgement as to the statistical rigor of the analyses. I suggested a number of corrections hence my selecting partly on whether the manuscript is technically sound.

Overall comment: In this study, the authors describe the prevalence and examine the sociodemographic correlates of accelerometer measured physical activity levels among school-going children in Kampala, Uganda. Using a multistage random sampling method, the authors recruited 256 participants from 7 primary schools. In my view, this is an important study which is timely and provides much needed objective data on childhood physical activity in a LMIC. The school environment plays an important if not a very significant role in children’s physical activity behaviors. I have identified important issues that the authors should consider making to improve their manuscript. Some parts of the methods should be expanded and made clear for the readers. Please consider having another co-author read through your manuscript for editorial reasons. There are several small but meaningful errors such as acronyms and wrong brackets for references that should be addressed

Introduction

1. Once you have defined Physical activity as PA, Moderate-to-Vigorous Intensity Physical activity as MVPA etc., after the first time you use the acronyms, try to be consistent and do not revert to spelling out the whole word and vice versa. It is something the authors should address throughout the manuscript. 

We thank the reviewer for pointing out this, it has been corrected throughout the revised version. We have changed PA into physical activity and used it throughout the manuscript.

2. Please add “for children” on line 59: after the phrase…revised their PA guidelines.

As requested we have added “for children” on line 69 in the revised version.

3. Line 62: after health benefits, could you please add something like “associated with regular participation in PA. 

As requested we have added “associated with regular participation in physical activity” (line 72 in the revised version).

4. Line 64: delete the word ‘also’ after Sallis and colleagues.

As requested we have deleted the word “also” after Sallis and colleagues (line 74 in the revised version)

5. Add a comma after the phrase …literature shows that in children, …

The whole sentence was removed because it was not supported by a review study and the current study did not focus on sedentary time.

6. References 13,14,15 on line 70, have different brackets to the rest of the references

We thank the reviewer for pointing out this and have put the square brackets (line 80 in the revised version).

7. The sentence starting on line 82 to 85 may need to be revised. Are you suggesting that all children have limited cognition, or do you mean limited reading comprehension skills due to their age? 

We meant limited reading and comprehension skills. This has been corrected in the revised version to read; Accelerometers are an alternative to self-report methods like questionnaires that are subject to recall bias and are not recommended for use among children because of their limited reading and comprehension skills due to their age [22,23]. (line 96/98)

8. When referring to children, it reads a bit odd to repeatedly say ‘in’ maybe consider using ‘among’

We have changed “in” to “among”, and used it throughout the manuscript.

9. References 13,26,31, line 100 and reference 35 on line 103 have different form of brackets

We thank the reviewer for pointing out this and have put the square brackets (references [13, 14, 15, 16] line 81 and reference [38] line 120 in the revised version).

10. Is HSES the same as SES? If so, please use the same acronym throughout your manuscript. Otherwise define HSES the first time you use it and then be consistent after that.

High socio economic status (HSES) is different from socioeconomic status (SES). We thank the reviewer for pointing this out, we have used the same acronym throughout the manuscript.

11. You use the acronym LICs on line 112, but I am not sure it has been defined before this mention

We have defined the acronym LICs on line 105 in the revised version and used it through out the manuscript.

12. Line 115, consider rewording to say: Therefore, the present study….

We have reworded as suggested on line 138 in the revised version

Methods

1. Consider starting the methods section with “This is a cross-sectional study of a representative sample of school-going children aged 10-12 years old in Kampala, Uganda”.

We thank the reviewer for pointing out this, it has been considered in the revised version on line 149.

2. Could you please substitute the word ‘tribes’ with perhaps ‘ethnic groups’? The word tribes has colonial connotations

We have substituted the word “tribe” with “ethnic group” on line 134 in the revised version.

3. How did you differentiate between awake non wear time and sleep time given that you are using a 24-hour protocol? Did you just lump possible sleep time and awake non wear time together because in your study you were not interested in measuring sleep duration? If so, please make that clear in your methods

We thank the reviewer for pointing out this, to address the reviewer’s concerns we have included information explaining this in the revised version (line 175/181) as shown here:

4. There is debate about the accuracy of classifying children over 5 years old as being ‘underweight’ rather than being ‘thin’ based on BMI z scores. Have authors thought The 24-hour protocol required sleep time to be identified and accounted for before evaluating wake wear time and generating physical activity variables of interest [51,52]. We used the Sadeh algorithm, which is in built into the sleep scoring function in ActiLife software to identify individualised daily sleep on set and offset time for each valid day for each child [53]; this is a valid method for removal of sleep [54]. Daily sleep on set and offset time was used to create time filters in CSV files (Excel Microsoft co-operation, 2016). Time filtered files for the wake period were created and used to identify non wear time and wear time.

5. about the implications of this give their reference to WHO 2007 BMI percentiles which uses thinness for the age group being studied here?

We have corrected this throughout the manuscript, however due to the small numbers of thin children we have combined them with normal weight to thin/normal weight. 

6. Was the questionnaire validated for this population? If so please make that clear in your description of the questionnaire.

We thank the reviewer for this important question. The questionnaire was validated among adults in Uganda. We have cleared this and included it in the description of the questionnaire on line 199 in the revised version. A validated questionnaire assessing children and parents’ socio-demographics and neighbourhood built environment [58] was completed by parents/guardians

7. Please clarify your recruitment and response rate. Are you calculating the percentages of children with consent out of 400 or the 600? If you are calculating it from 600 then your percentages are off. The first one 66.7% is accurate but the next two need to be described accurately. For example, you could say “of the 600 who were invited to participate, 400 (66.7%) had parents who completed questionnaires…etc.

We have corrected this to read; Of the 600 children who were invited to participate, 400 (66.7%) had parents/guardians who completed the questionnaire and 328 (54.6%) parental/guardian consented for their children to participate in accelerometry and anthropometric assessment. Of the 328 children who obtained parental consent to wear devices, 256 (78%) had valid accelerometry data and were therefore retained for analysis. (Line 216/223 in the revised version).

8. Was there a specific reason for authors to use bivariate logistic regression as opposed to some hierarchical model which would be more robust to account for the clustering at the school and classroom levels?

We used a Multilevel mixed-effects logistic regression where we adjusted for both school and division (line 231/232 in the revised version). This accounted for the hierarchical element in the data.

melogit outcome exposure || division || school:, or

Results

1. Generally, it may be better to avoid the use or terms such as ‘almost’ and rather use approximately, about, close to…etc

We thank the reviewer for pointing out this, it has been considered throughout the manuscript.

2. Be consistent, either use numbers or words when writing numerical data in text. For example ‘One in every 5 parents/guardians’…may be better to just say one in every five…

We have corrected this in the revised version (Line 262)

3. Do you mean approximately ‘equal numbers by age’? You have three ages and as such could not be ‘even’.

We thank the reviewer for pointing this out we have corrected it to read; Most of the children were aged 10 and 11 years old (71.5%). (line 258 in the revised version)

4. Table 1: Overweight is one word.

We have corrected it; however due to the small numbers we have combined overweight and obese children.

5. Consider using the ‘merge cells’ function to make your sub-heading lines one line in tables. 

We have merged the cells as suggested throughout the manuscript.

6. You are using children/participants interchangeably in your results section. Could you please pick one and be consistent with it.

We thank the reviewer for pointing out this inconsistency, we have used the word children throughout the manuscript.

7. Line 230: ‘Overall on average the children spent...’ seem awkward... can you consider revising for readability?

We have revised the sentence to read; Most of the families studied (62.1%) had 2 to 4 children aged 0 to 17 years old. (line 275) revised version

8. The sentence on lines 251 starting with ‘Males…’ is awkward, consider rephrasing it.

We have rephrased the sentence to read; Significantly more males (38.9%) than females (34.3%) accumulated recommended volumes of MVPA. (line 299 revised version

Discussion

1. Do you think that it was overweight/obesity that caused the participants to be more sedentary or that it was the sedentariness that led to being overweight/obese? Reverse causality is a possibility isn’t it? What is your comment?

The current study was not a causality study, we only sought to determine possible associations. However, it is true that reverse causality is highly possible and also the two aspects can still exist in a population. Therefore, future studies are required to further examine this. (also see comment in the revised version line 319).

Generally, well done, consider revising some statements which are awkwardly worded throughout the discussion.

We thank the reviewer for pointing out this, we have revised the awkwardly worded statements throughout the discussion as suggested.

---

## [Decision Letter · Decision Letter 1]

4 May 2020

PONE-D-20-00496R1

Prevalence and sociodemographic correlates of accelerometer measured physical activity levels of school-going children in Kampala city, Uganda.

PLOS ONE

Dear Ms Nakabazzi,

Thank you for submitting your manuscript to PLOS ONE. After careful consideration, we feel that it has merit but does not fully meet PLOS ONE’s publication criteria as it currently stands. Therefore, we invite you to submit a revised version of the manuscript that addresses the points raised during the review process.

I would like to congratulate the author for addressing the reviewers' comments. However, there are minor issues to review and I encourage the authors to review with detail the text to avoid spell mistakes.

We would appreciate receiving your revised manuscript by Jun 18 2020 11:59PM. To enhance the reproducibility of your results, we recommend that if applicable you deposit your laboratory protocols in protocols.io, where a protocol can be assigned its own identifier (DOI) such that it can be cited independently in the future. For instructions see: http://journals.plos.org/plosone/s/submission-guidelines#loc-laboratory-protocols

We look forward to receiving your revised manuscript.

Kind regards,

Javier Brazo-Sayavera, Ph.D.

Academic Editor

PLOS ONE

Reviewers' comments:

Reviewer's Responses to Questions

**Comments to the Author**

1. If the authors have adequately addressed your comments raised in a previous round of review and you feel that this manuscript is now acceptable for publication, you may indicate that here to bypass the “Comments to the Author” section, enter your conflict of interest statement in the “Confidential to Editor” section, and submit your "Accept" recommendation.

Reviewer #1: All comments have been addressed

Reviewer #2: All comments have been addressed

2. Is the manuscript technically sound, and do the data support the conclusions?

Reviewer #1: Yes

Reviewer #2: Yes

3. Has the statistical analysis been performed appropriately and rigorously? 

Reviewer #1: Yes

Reviewer #2: Yes

4. Have the authors made all data underlying the findings in their manuscript fully available?

Reviewer #1: Yes

Reviewer #2: Yes

5. Is the manuscript presented in an intelligible fashion and written in standard English?

Reviewer #1: Yes

Reviewer #2: Yes

6. Review Comments to the Author

Reviewer #1: Very well done with addressing my comments. There are some minor typos, grammatical errors, and issues with the use of uppercase and lowercase letters. I'm sure the editorial office will pick these issues up during copy editing and proofreading.

I have (almost) no further comments; just a few suggestions for the authors to consider:

1. If possible, avoid the use of "inactive" or "inactivity", instead, use "insufficiently active" or "insufficient activity". A healthy person can not be inactive, I guess.

2. Please clarify the response rate issue - clearly state "The response rate was XX%". Consider adding a few sentences about a seemingly low-response rate in the discussion (limitations) section.

3. In the Discussion section, please comment on the policy implications of these findings. How the results can be instrumental in informing active lifestyle strategies in Uganda, what policy initiatives may be required, how the government can align their efforts with the global community, e.g., WHO, Active Healthy Kids Global Alliance. The authors may wish to read Aubert et al. (2018) Global Matrix 3.0 Physical activity report card grades for children and youth: Results and analysis from 49 countries. Journal of Physical Activity and Health 15 (Supplement 2), S251-S273. I would encourage the authors to add a dedicated paragraph on "Implications for future research and policy" in the Discussion section before the Limitations of the study (no subheading required).

Reviewer #2: The authors have sufficiently addressed all comments and questions I raised. I have no further questions nor concerns with this manuscript.

7. PLOS authors have the option to publish the peer review history of their article (what does this mean?). If published, this will include your full peer review and any attached files.

Reviewer #1: No

Reviewer #2: No

---

## [Author Response · Author response to Decision Letter 1]

10 Jun 2020

Academic Editor:

No comments provided 

Reviewer #1

A. Very well done with addressing my comments. There are some minor typos, grammatical errors, and issues with the use of uppercase and lowercase letters. I'm sure the editorial office will pick these issues up during copy editing and proofreading.

We thank the reviewer for pointing out this, we have addressed the typing and grammatical errors, and the use of upper case and lower case throughout the manuscript.

B. I have (almost) no further comments; just a few suggestions for the authors to consider:

1. If possible, avoid the use of "inactive" or "inactivity", instead, use "insufficiently active" or "insufficient activity". A healthy person cannot be inactive, I guess.

We thank the reviewer for this suggestion, it has been corrected throughout the revised version. We have changed the words “inactive or Inactivity” to “insufficiently active or insufficient activity” and used them throughout the manuscript.

2. Please clarify the response rate issue - clearly state "The response rate was XX%". Consider adding a few sentences about a seemingly low-response rate in the discussion (limitations) section.

We thank the reviewer for pointing this out we have clarified the response rate issue and clearly stated it as; “The response rate was 42.7% (line 199-200 in the revised version). Also the low response rate has been included in the study limitations (line 398 in the revised version).

3. In the Discussion section, please comment on the policy implications of these findings. How the results can be instrumental in informing active lifestyle strategies in Uganda, what policy initiatives may be required, how the government can align their efforts with the global community, e.g., WHO, Active Healthy Kids Global Alliance. The authors may wish to read Aubert et al. (2018) Global Matrix 3.0 Physical activity report card grades for children and youth: Results and analysis from 49 countries. Journal of Physical Activity and Health 15 (Supplement 2), S251-S273. I would encourage the authors to add a dedicated paragraph on "Implications for future research and policy" in the Discussion section before the Limitations of the study (no subheading required).

We thank the reviewer for suggesting this very important paper, we have included a paragraph in the discussion section on strategies to increase physical activity and policy implications of the current study results to read as;

“Therefore, there is need for developing effective strategies and policies with the aim of increasing physical activity levels among school going children in Kampala city and Uganda. This may be achieved by implementing strategies and policies that have been proposed by various global and regional organisations including those of the Active Healthy Kids Global Alliance (AHKGA) in the fight against the insufficient physical activity among children [81,82]. The current study further highlighted the need for nationally representative physical activity data. The Ministry of Education and Sports in Uganda should fund the development and release of a national report card on physical activity for children in Uganda for surveillance and promotion of physical activity among Ugandan children”.

Reviewer # 2

The authors have sufficiently addressed all comments and questions I raised. I have no further questions nor concerns with this manuscript.

---

## [Decision Letter · Decision Letter 2]

11 Jun 2020

Prevalence and sociodemographic correlates of accelerometer measured physical activity levels of school-going children in Kampala city, Uganda.

PONE-D-20-00496R2

Dear Dr. Nakabazzi,

We’re pleased to inform you that your manuscript has been judged scientifically suitable for publication and will be formally accepted for publication once it meets all outstanding technical requirements.

Kind regards,

Javier Brazo-Sayavera, Ph.D.

Academic Editor

PLOS ONE

Reviewer's Responses to Questions

**Comments to the Author**

1. If the authors have adequately addressed your comments raised in a previous round of review and you feel that this manuscript is now acceptable for publication, you may indicate that here to bypass the “Comments to the Author” section, enter your conflict of interest statement in the “Confidential to Editor” section, and submit your "Accept" recommendation.

Reviewer #1: All comments have been addressed

2. Is the manuscript technically sound, and do the data support the conclusions?

Reviewer #1: Yes

3. Has the statistical analysis been performed appropriately and rigorously? 

Reviewer #1: Yes

4. Have the authors made all data underlying the findings in their manuscript fully available?

Reviewer #1: Yes

5. Is the manuscript presented in an intelligible fashion and written in standard English?

Reviewer #1: Yes

6. Review Comments to the Author

Reviewer #1: No further comments for the authors. The authors have done a wonderful job addressing my comments on the previous version of the manuscript.

7. PLOS authors have the option to publish the peer review history of their article (what does this mean?). If published, this will include your full peer review and any attached files.

Reviewer #1: No

---

## [Editor Report · Acceptance letter]

29 Jun 2020

PONE-D-20-00496R2 

Prevalence and socio-demographic correlates of accelerometer measured physical activity levels of school-going children in Kampala city, Uganda 

Dear Dr. Nakabazzi:

I'm pleased to inform you that your manuscript has been deemed suitable for publication in PLOS ONE. Congratulations! Your manuscript is now with our production department. 

Kind regards, 

on behalf of

Dr. Javier Brazo-Sayavera 

Academic Editor

PLOS ONE